# AlphaQCM: Alpha Discovery with Distributional Reinforcement Learning

## Abstract

Finding synergistic formulaic alphas is very important but challenging for researchers and practitioners in finance. In this paper, we reconsider the discovery of formulaic alphas from the viewpoint of sequential decision-making, and conceptualize the entire alpha discover process as a non-stationary and reward-sparse Markov decision process. To overcome the challenges of non-stationarity and reward-sparsity, we propose the AlphaQCM method, a novel distributional reinforcement learning method designed to search for synergistic formulaic alphas efficiently. The AlphaQCM method first learns the Q function and quantiles via a Q network and a quantile network, respectively. Then, the AlphaQCM method applies the quantiled conditional moment method to learn unbiased variance from the potentially biased quantiles. Guided by the learned Q function and variance, the AlphaQCM method navigates the non-stationarity and reward-sparsity to explore the vast search space of formulaic alphas with high efficacy. Empirical applications to real-world datasets demonstrate that our AlphaQCM method significantly outperforms its competitors, particularly when dealing with large datasets comprising numerous stocks.

## 1 Introduction

Over the past decades, extensive research has investigated the predictive power of historical stock information for forecasting future returns, resulting in the development of several well-known alphas. Here, each alpha is a function that transforms noisy historical stock data into signals for predicting future stock returns. However, recent studies examining the influence of investor behavior and psychology on price dynamics have uncovered the existence of subtle and intricate alphas that are difficult to formalize using standard financial methods (Barberis, 2018).

Most existing AI approaches surpass traditional methods in this field by designing alphas through the use of sophisticated machine learning (ML) models in an end-to-end manner (Feng et al., 2019; Ding et al., 2020; Koa et al., 2023). Intuitively, the well-trained ML models are effective alphas, since they can transform stock data into predictive signals. However, they are inherently complex and lack simple mathematical representations, leading to the so-called non-formulaic alphas. Yet, the non-formulaic alphas have trust issues due to their black-box nature, so they are not widely adopted in the industry.

For dealing with the above trust issue, emerging literature focuses on **how to automatically discover a set of synergistic formulaic alphas**. The formulaic nature of these alphas usually makes them compact, present explicit interpretations, and generalize well; meanwhile, their synergistic nature allows them to be combined into a meta-alpha via some interpretable models (e.g., linear models). Finding these formulaic alphas typically is based on the genetic programming (GP) method (Lin et al., 2019a;b; Zhang et al., 2020; Cui et al., 2021). However, the GP method has a vast search space, which scales exponentially with the number of input features and operators.

To overcome the challenge of vast search space in the GP method, the AlphaGen method (Yu et al., 2023) reformulates the alpha discovery problem into the task of finding an optimal policy for a specialized Markov decision process (MDP), and then achieves this task via a reinforcement learning (RL) algorithm. Although the AlphaGen method shows state-of-the-art performance, it has two substantial theory-practice gaps, resulting in the inefficient and unstable alpha discovery procedure in practice. The first gap is an unaddressed non-stationary issue (Lecarpentier & Rachelson, 2019),

where the reward function in the MDP of interest changes dynamically across decision epochs. This challenge stems from the objective of identifying a set of synergistic formulaic alphas rather than a single formulaic alpha. This issue arises from the goal of collecting a set of synergistic formulaic alphas rather than a single formulaic alpha. The second gap is how to accommodate the reward-sparse nature of the considered MDP, as most discovered alphas are weak and leads to zero rewards.

This paper contributes to the literature by introducing a novel method, AlphaQCM, which addresses these theory-practice gaps and presents a new RL solution to the alpha discovery problem. Specifically, the AlphaQCM method first leverages the IQN algorithm (Dabney et al., 2018a), to learn the quantiles of cumulative discounted rewards, whereas it studies the mean of cumulative discounted rewards via the DQN algorithm (Mnih et al., 2015). Then, based on the learned quantiles, the AlphaQCM method adopts the quantiled conditional moments (QCM) method (Zhang & Zhu, 2023) to estimate variance of cumulative discounted rewards, which serves as a natural exploration bonus for the mining agent's action selection to relieve the issue of reward-sparsity. Remarkably, the estimated variance from the QCM method remains unbiased even if the estimated quantiles are biased due to non-stationarity. Hence, by employing the AlphaQCM method, we can alleviate the negative impacts of non-stationarity and reward-sparsity, thereby achieving a significantly better empirical performance in discovering formulaic alphas. Our work clearly generalizes and extends the Alpha-Gen method and can be applied to other non-stationary and/or reward-sparse environments.

We apply our AlphaQCM method to three real-world market datasets to assess its empirical performance, together with baseline methods such as the AlphaGen method and GP-based methods. Extensive experimental results demonstrate that the AlphaQCM method consistently achieves the best performance, with Information Coefficient (IC) values of $8.49\%$, $9.55\%$, and $9.16\%$ across the three datasets. Its superior performance is particularly evident when the dataset originates from a complex financial system. Finally, we conduct several ablation studies to investigate the contribution of each component in the AlphaQCM method.

## 2 BACKGROUND AND RELATED WORK

### 2.1 FORMULAIC ALPHA

The formulaic alpha has an extensive search space of potential expressions due to the enormous operators and features that are available for selection. Generally speaking, most existing methods for discovering formulaic alphas can be categorized into two classes: GP-based methods and RL-based methods. In the past decade, the GP-based methods have predominantly served as the mainstream to generate formulaic alphas (Lin et al., 2019a;b; Zhang et al., 2020; Cui et al., 2021). For example, the AlphaEvolve method (Cui et al., 2021) evolves new alphas from existing ones using the AutoML-Zero framework (Real et al., 2020), with the IC employed as the fitness measure. However, the recent literature highlights the suboptimal performance of GP-based methods in scenarios involving large populations (Petersen et al., 2021), which are essential for alpha discovery due to the complexity of considered financial market.

Conversely, as a RL-based method, the AlphaGen method (Yu et al., 2023) conceptualizes the alpha discovery process as a Markov decision process (MDP) and employs an RL algorithm, specifically the proximal policy optimization (PPO) algorithm (Schulman et al., 2017), to discover a set of synergistic alphas with high returns. Although the AlphaGen method has significantly outperformed the previous GP-based methods, it has three notable shortcomings. First, due to the reward-sparse nature of the alpha discovery MDP, the AlphaGen method struggles to explore the search space efficiently. Second, the AlphaGen method suffers from the issues related to sample efficiency and convergence performance, as the alpha discovery MDP is clearly non-stationary. Third, the AlphaGen method completely ignores the intricate distributional information within the observed expressions and subsequent alpha construction, resulting in an inefficient and unstable alpha discovery process.

### 2.2 DISTRIBUTIONAL REINFORCEMENT LEARNING

Following the standard RL setting, the agent-environment interactions are modeled as an MDP, $(\mathcal{X}, \mathcal{A}, \mathcal{P}, \gamma, \mathcal{R})$, where $\mathcal{X}$ and $\mathcal{A}$ are finite sets of states and actions, respectively, $\mathcal{P} : \mathcal{X} \times \mathcal{A} \to \mathscr{P}(\mathcal{X})$ is the transition kernel, $\gamma \in [0, 1)$ is the discount factor, and $\mathcal{R} : \mathcal{X} \times \mathcal{A} \to \mathscr{P}(\mathbb{R})$ is the reward function with $\mathscr{P}(A)$ being a random variable with support $A$.

At the $t$-th agent-environment interaction, the agent observes state $X_t \sim \mathscr{P}(\mathcal{X})$, selects action $A_t \sim \mathscr{P}(\mathcal{A})$, and subsequently receives feedback from the environment in the form of the next state $X_{t+1} \sim \mathcal{P}(\cdot \mid X_t, A_t)$ and reward $R_t \sim \mathcal{R}(X_t, A_t)$. For a given policy $\pi : \mathcal{X} \to \mathscr{P}(\mathcal{A})$, the discounted cumulative reward can be represented by a random variable $Z^\pi(x, a)$:

$$Z^\pi(x,a) = \sum_{t=0}^{\infty} \gamma^t [\mathcal{R}(X_t, A_t) \mid X_0 = x, A_0 = a] = \sum_{t=0}^{\infty} \gamma^t (R_t \mid X_0 = x, A_0 = a),$$

where $A_t \sim \pi(\cdot | X_t)$, and $(x, a) \in \mathcal{X} \times \mathcal{A}$ is a state-action pair[1].

The ultimate goal of RL algorithms is to ascertain an optimal policy $\pi^*$, which maximizes the expectation of discounted cumulative rewards, also known as the Q function $Q^\pi(x, a) \equiv \mathbb{E}[Z^\pi(x, a)]$. A common way to obtain $\pi^*$ is to find the unique fixed point $Q^* \equiv Q^{\pi^*}$ of the Bellman optimality operator $\mathcal{T}$ (Bellman, 1966), satisfying

$$Q^*(x,a) = \mathcal{T}Q^*(x,a) \equiv \mathbb{E}\left[ R_t + \gamma \max_{a' \in \mathcal{A}} Q^*(X_{t+1}, a') \mid X_t = x, A_t = a \right].$$

In practice, $Q^*$ is typically approximated by a parametric function, such as the deep Q network (Mnih et al., 2015). However, the majority of RL algorithms only focus on the scalar expectation $Q^*(x, a)$, thereby overlooking the valuable distributional information arising from the potential randomness of optimal policy and the stochasticity of considered environment.

To address this issue, the distributional RL (DRL) algorithms concentrate on directly learning the distribution of $Z^\pi(x, a)$. Let $Z^* \equiv Z^{\pi^*}$ denote the discounted cumulative rewards with optimal policy $\pi^*$. To find $\pi^*$, the distributional Bellman optimality operator $\mathcal{T}^D$ (Bellemare et al., 2017) is defined as:

$$Z^*(x,a) = \mathcal{T}^D Z^*(x,a) \overset{D}{=} R_t + \gamma Z^*(X_{t+1}, a') \mid X_t = x, A_t = a, \tag{1}$$

where $a' = \arg\max_{a \in \mathcal{A}} \mathbb{E}[Z^*(X_{t+1}, a)]$ and $\overset{D}{=}$ denotes the equality in probability laws.

In practice, it is common to parameterize $Z^*$ via the quantile representation $Z_{\boldsymbol{\theta}, \boldsymbol{\tau}}(x, a)$ (Dabney et al., 2018b;a; Hessel et al., 2018; Yang et al., 2019), which is a mixture of $K$ Dirac distributions. Specifically,

$$Z_{\boldsymbol{\theta}, \boldsymbol{\tau}}(x,a) = \sum_{k=0}^{K-1} (\tau_{k+1} - \tau_k)\delta_{\theta_k(x,a)}, \tag{2}$$

where $\delta_z$ is a Dirac distribution centered at $z \in \mathbb{R}$, $\boldsymbol{\tau} = (\tau_1, \ldots, \tau_K)'$ is a vector of quantile levels satisfying $0 < \tau_1 < \cdots < \tau_{K-1} < \tau_K = 1$, and $\boldsymbol{\theta} = \{\theta_1, \ldots, \theta_K\}$ is a set of functions. Moreover, we denote $\boldsymbol{\theta}(x, a) = (\theta_1(x, a), \ldots, \theta_K(x, a))' \in \mathbb{R}^K$, where $\theta_k(x, a)$ is the $\tau_k^*$-th quantile of $Z^*(x, a)$ with $\tau_k^* = (\tau_{k+1} + \tau_k)/2$ and $\tau_0 = 0$.

Needless to say, the formulation of $Z_{\boldsymbol{\theta}, \boldsymbol{\tau}}(x, a)$ depends on $\boldsymbol{\tau}$ and $\boldsymbol{\theta}$. As a pioneer work, the QRDQN algorithm (Dabney et al., 2018b) adopts a fixed $\boldsymbol{\tau}$, and applies a multi-head deep neural network to learn corresponding $\theta_k(x, a)$. Furthermore, the IQN algorithm (Dabney et al., 2018a) considers a random $\boldsymbol{\tau}$, while the FQF algorithm (Yang et al., 2019) incorporates an additional neural network to learn the optimal $\boldsymbol{\tau}$ for each $(x, a)$.

Accompanied by the quantile representation, the existing DRL algorithms first learn $\boldsymbol{\theta}(x, a)$, and then estimate $Q^*(x, a)$ by directly taking the expectation of $Z_{\boldsymbol{\theta}, \boldsymbol{\tau}}(x, a)$. Although the empirical evidence suggests that DRL algorithms have the good performance with the desirable robustness to variations in hyperparameters, several challenges remain. The most critical challenge is the validity of the estimated quantiles, raising a concern about the consistency of $Z_{\boldsymbol{\theta}, \boldsymbol{\tau}}(x, a)$ and its moments. Specifically, the traditional DRL algorithms show the unsatisfactory performance in non-stationary and reward-sparse MDPs, which are common in practical scenarios (e.g., the alpha discovery MDP discussed in this paper).

---

[1]In this paper, random variables are represented by uppercase letters (e.g., $X_t$) and observations are represented by lowercase letters (e.g., $x_t$).

## 3 METHODOLOGY

In this paper, we consider a scenario involving $N$ distinct stocks with their prices and volume information. Our goal is to find an optimal alpha pool $\mathcal{F}$ (i.e., a set of synergistic formulaic alphas), which is effective for constructing a predictive linear meta-alpha for future stock returns.

To be more specific, we assume that the alpha pool $\mathcal{F}$ comprises at most $P$ different formulaic alphas, labeled as $f_1, \ldots, f_P$. For $p = 1, \ldots, P$, each $f_p$ is a function that maps market data into alpha values, defined as follows:

$$\boldsymbol{\alpha}_{p,s} = f_p(\boldsymbol{H}_{s-1}) \in \mathbb{R}^N,$$

where $\boldsymbol{H}_{s-1}$ includes historical information for $N$ stocks up to time $s-1$, and $\boldsymbol{\alpha}_{p,s}$ is the vector of cross-sectional alpha values. Then, the linear meta-alpha $\widehat{\boldsymbol{\alpha}}_s$ is defined as:

$$\widehat{\boldsymbol{\alpha}}_s = \sum_{p=1}^{P} \boldsymbol{\alpha}_{p,s} \widehat{\beta}_p \in \mathbb{R}^N,$$

where $\widehat{\beta}_p \in \mathbb{R}$ for $p = 1, \ldots, P$ is the linear coefficient (or weight) for each alpha, and $\widehat{\boldsymbol{\alpha}}_s$ are expected to effectively capture subsequent stock returns $\boldsymbol{y}_s \in \mathbb{R}^N$, thereby exhibiting a high IC value[2]. To achieve this goal, we conceptualize the alpha discovery process as a non-stationary and reward-sparse MDP and propose the AlphaQCM method, a novel and efficient DRL algorithm, to find the most effective $\mathcal{F}$.

### 3.1 REPRESENTATION OF FORMULAIC ALPHA

Before going further, we begin by introducing how to reformulate the alpha discovery problem into a sequential decision-making task. Recall that each formulaic alpha has a mathematical expression comprising operators and features. These operators are broadly categorized into time-series operators and cross-sectional operators (Kakushadze, 2016; Lin et al., 2019a;b). The time-series operators necessitate data spanning multiple days, such as *TsRank(Close, d)*, which calculates the sequential ranking of the most recent $d$ closing prices for each stock. In contrast, the cross-sectional operators manipulate single-day data, exemplified by *Rank(Low)*, which computes the cross-sectional ranking of low prices among $N$ stocks. By combining these two types of operators, the formulaic alphas exhibit high nonlinearity, but are still interpretable for humans.

After specifying the operators used in the mathematical expression, we further employ the Reverse Polish Notation (RPN) method to represent the expression (i.e., the form of $f_p$). Figure 1 gives a specific illustrating example on the RPN method. In this figure, the formulaic alpha "Alpha#4" is encoded into a token sequence, where *BEG* and *SEP* tokens indicate the beginning and ending of the expression, respectively, and each feature or operator is denoted as a token. The details of the available features and operators are listed in Table B.3.

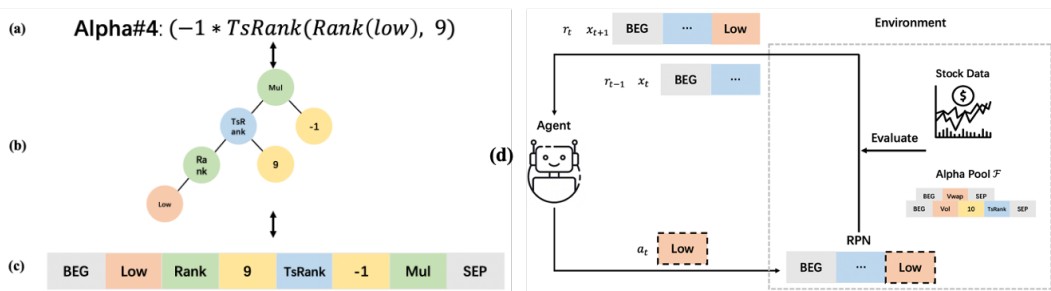

Figure 1: (a) Formulaic expression of Alpha#4 factor in (Kakushadze, 2016). (b) Its expression tree. (c) Its RPN representation. (d) Agent-environment interaction diagram for alpha discovery.

Intuitively, the RPN representation can be viewed as a trajectory, documenting the sequential actions taken by an agent, when selecting which token to place at each position of formula. As a result, the

---

[2]See Appendix A for more explanations.

task of discovering formulaic alphas can be regarded as a sequential decision-making problem by designing a particular MDP.

## 3.2 SPECIFICATION OF MDP

After introducing the RPN representation, we specify the alpha discovery MDP, as illustrated in Figure 1(d). Below, we show each component in this MDP.

**State and action.** In accordance with the GP-based methods, the agent manipulates token sequences, prompting us to consider a token-based state set $\mathcal{X}$. Specifically, each state observation $x_t \in \mathcal{X}$ corresponds to a token sequence representing the currently generated expression, with initial state $X_0 \equiv x_0$ being the *BEG* token. To maintain the interpretability of discovered alphas, we restrict any state to have fewer than 20 tokens. Aligned with the design of state, each action $a_t \in \mathcal{A}$ is a token. However, only a subset of $\mathcal{A}$ is allowed to be taken for a specific $x_t$[3], since not all token sequence are guaranteed to be the RPNs of valid formulaic alphas.

**Transition kernel.** In this MDP, the transition kernel is deterministic. Given $x_t$ and $a_t$, the environment feedbacks the next state $x_{t+1}$ by appending $a_t$ to the end of $x_t$, unless $a_t$ is the *SEP* token or $x_t$ has reached its maximum length. In such cases, this episode of agent-environment interaction terminates, and the environment presents $x_0$ to the agent to initiate a new episode.

**Reward.** The key component of the alpha discovery process is the design of reward. Intuitively, the reward is expected to quantify the contribution of the newly discovered formulaic alpha on the current alpha pool $\mathcal{F}$, which consists of at most $P$ different formulaic alphas collected in previous episodes. Following the idea outlined in Yu et al. (2023), we set $r_t = 0$ for any incomplete token sequence, as the formulaic alpha is only partially formed. Once the token sequence is completed, $x_t$ is parsed into a formulaic alpha. If the parsed alpha is invalid, $r_t = -1$; otherwise, the formulaic alpha is evaluated by the environment through the following steps: (1) The new formulaic alpha is added to alpha pool to create an extended alpha pool; (2) A linear model is fitted based on the extended alpha pool to select up to $P$ principal formulaic alphas with the most significant contribution, and the alpha pool is updated accordingly; (3) The meta-alpha is obtained based on the updated alpha pool and the fitted linear model; (4) The reward $r_t$ is calculated by the increase in IC of the meta-alpha based on the updated alpha pool compared to that based on previous alpha pool. See Algorithm C.1 for more details.

**Discount factor.** Although long alphas tend to lack generalizability and interpretability, they often are more sophisticated and predictive for stock trends. While the max length of episode is restricted, we hence set $\gamma = 1$.

**Non-stationarity.** Since the reward function varies across epochs (with different formulaic alphas collected by the alpha pool), the alpha discovery MDP is clearly non-stationary. Specifically, when the agent discovers a new formulaic alpha that proves to be effective (as indicated by a high reward from the environment), the environment incorporates this formulaic alpha into the alpha pool. In the subsequent episode, the similar formulaic alphas to the recently collected one tend to earn little reward, since the alpha pool has been updated to capture the predictive information related to this type of alpha. As a result, once the alpha pool is updated, the reward function of the MDP changes, necessitating numerous agent-environment interactions and extensive training to re-learn it.

**Reward-sparsity.** In addition to the non-stationarity, the alpha discovery MDP exhibits reward-sparsity from two perspectives. First, the reward can only be non-zero when the episode ends (i.e., when a new formulaic alpha is generated); otherwise, it must be zero. Second, due to the low signal-to-noise ratio of market datasets, most discovered alphas are meaningless and not beneficial to the alpha pool, leading to zero rewards. Consequently, there are numerous zero rewards in the transitions, resulting in an inefficient and unstable training process for the agent.

## 3.3 ALPHAQCM

To solve the issues of non-stationarity and reward-sparsity in alpha discovery MDP, our AlphaQCM method employs the QCM method (Zhang & Zhu, 2023) to learn an unbiased variance of rewards.

---

[3]For example, a time delta token must be followed by a time-series operator. See Appendix C of Yu et al. (2023) for more details on this aspect.

This variance is further used as a bonus to guide the agent in exploring the environment, thereby improving the agent's training efficiency. Below, we outline the core modules of our AlphaQCM method.

### 3.3.1 QCM

We begin with introducing how to learn variance from quantiles via the QCM method, which can be extended to most existing DRL frameworks. Recall that the goal of a DRL algorithm is to model the discounted cumulative reward $Z^*(x, a)$ with its quantiles $\theta_k(x, a)$ for $k = 1, \ldots, K$, as in (2). According to the Cornish-Fisher expansion (Cornish & Fisher, 1938), the quantiles of $Z^*(x, a)$ are linked to its moments as follows:

$$
\begin{aligned}
\theta_k(x, a) = Q^*(x, a) + z_k \sqrt{h(x, a)} + \left(z_k^2 - 1\right) \frac{\sqrt{h(x, a)} s(x, a)}{6} \\
+ \left(z_k^3 - 3 z_k\right) \frac{\sqrt{h(x, a)} \left[k(x, a) - 3\right]}{24} + \omega_k(x, a),
\end{aligned}
\tag{3}
$$

where $z_k$ is the $\tau_k^*$-th quantile of Gaussian distribution, $Q^*(x, a)$, $h(x, a)$, $s(x, a)$, and $k(x, a)$ denote the mean, variance, skewness, and kurtosis of $Z^*(x, a)$, respectively, and $\omega_k(x, a)$ represents the remaining term of this expansion.

The estimated quantile $\widehat{\theta}_k(x, a)$ is expected to oscillate around $\theta_k(x, a)$, but it is always biased due to the presence of non-stationarity. Specifically, we model this phenomenon as $\widehat{\theta}_k(x, a) = \theta_k(x, a) + \zeta_k^\circ(x, a) + \varepsilon_k^\circ(x, a)$, where $\zeta_k^\circ(x, a)$ is an unknown bias term, and $\varepsilon_k^\circ(x, a)$ is a zero-mean error term. Clearly, $\zeta^\circ(x, a)$ would be far away from zero when the alpha pool is newly updated.

By substituting $\theta_k(x, a)$ with $\widehat{\theta}_k(x, a)$ in (3), we have

$$
\begin{aligned}
\widehat{\theta}_k(x, a) = \zeta(x, a) + Q^*(x, a) + z_k \sqrt{h(x, a)} + \left(z_k^2 - 1\right) \frac{\sqrt{h(x, a)} s(x, a)}{6} \\
+ \left(z_k^3 - 3 z_k\right) \frac{\sqrt{h(x, a)} \left[k(x, a) - 3\right]}{24} + \varepsilon_k(x, a),
\end{aligned}
\tag{4}
$$

where $\varepsilon_k(x, a) = \varepsilon_k^\bullet(x, a) - \zeta(x, a)$ with $\varepsilon_k^\bullet(x, a) = \zeta_k^\circ(x, a) + \varepsilon_k^\circ(x, a) + \omega_k(x, a)$ and $\zeta(x, a) = \mathbb{E}[\varepsilon_k^\bullet(x, a)]$. Intuitively, $\zeta(x, a)$ encompasses all biases caused by estimation and expansion.

Next, by gathering the $K$ quantiles together, we construct the following linear regression model:

$$
\begin{pmatrix} \widehat{\theta}_1(x, a) \\ \widehat{\theta}_2(x, a) \\ \vdots \\ \widehat{\theta}_K(x, a) \end{pmatrix} = \begin{pmatrix} 1 & z_1 & z_1^2 - 1 & z_1^3 - 3 z_1 \\ 1 & z_2 & z_2^2 - 1 & z_2^3 - 3 z_2 \\ \vdots & \vdots & \vdots & \vdots \\ 1 & z_K & z_K^2 - 1 & z_K^3 - 3 z_K \end{pmatrix} \begin{pmatrix} \zeta(x, a) + Q^*(x, a) \\ \sqrt{h(x, a)} \\ \frac{\sqrt{h(x, a)} s(x, a)}{6} \\ \frac{\sqrt{h(x, a)} [k(x, a) - 3]}{24} \end{pmatrix} + \begin{pmatrix} \varepsilon_1(x, a) \\ \varepsilon_2(x, a) \\ \vdots \\ \varepsilon_K(x, a) \end{pmatrix}.
$$

By solving the above linear regression, we can obtain the estimators $\widehat{h}(x, a)$, $\widehat{s}(x, a)$, and $\widehat{k}(x, a)$. Under some mild conditions[4], the consistency of these moment estimators is guaranteed by:

**Proposition 3.1.** *Suppose that Assumptions 1 and 2 hold. Then, $\widehat{h}(x, a) \xrightarrow{p} h(x, a)$, $\widehat{s}(x, a) \xrightarrow{p} s(x, a)$ and $\widehat{k}(x, a) \xrightarrow{p} k(x, a)$ as $K \to \infty$, where $\xrightarrow{p}$ denotes convergence in probability.*

Although the MDP is non-stationary, $\widehat{h}(x, a)$ remains unbiased, whereas there is no such guarantee for the vanilla quantile-based variance estimator[5], even in stationary MDPs (Bellemare et al., 2023). As mentioned in Mavrin et al. (2019), $\widehat{h}(x, a)$ captures both parametric and intrinsic uncertainties, which can be attributed to non-stationarity and reward-sparsity, respectively. Using $\widehat{h}(x, a)$ as an exploration bonus, our agent tends to explore the most uncertain states, which also lead to the most informative experiences for overcoming the challenges of non-stationarity and reward-sparsity. By training on these informative experiences, the agent mitigates the negative impacts of reward sparsity and non-stationarity as much as possible and efficiently learns from the dynamic environment.

---

[4]See Appendix D for more details.

[5]The vanilla quantile-based variance estimator is defined in Appendix E.

Lastly, it should be noted that we cannot estimate $Q^*(x, a)$ using the QCM method, since both $\zeta(x, a)$ and $Q^*(x, a)$ are unidentifiable in (4). Therefore, we use a separate RL algorithm to learn $Q^*(x, a)$. In such a non-stationary MDP, the traditional DRL algorithms yield biased Q estimates, as they estimate $Q^*(x, a)$ by directly taking expectation of $Z_{\widehat{\theta}, \tau}(x, a)$. This biased Q issue in non-stationary MDPs is somewhat inevitable, but using the QCM method can alleviate the negative impacts from non-stationarity. The underlying reason is that using QCM method enhances training efficiency, regardless of whether the bias caused by non-stationarity exists. By improving training efficiency, the agent requires fewer agent-environment interactions and less training time to re-approximate $Q^*(x, a)$.

### 3.3.2 DRL BACKBONE

After showing how to use the QCM method to obtain variance from quantiles, we elaborate on the backbone used to estimate the quantiles and Q function. In this paper, we adopt the IQN algorithm (Dabney et al., 2018a) to learn the quantiles, and apply the DQN algorithm (Mnih et al., 2015) to learn the Q function.

Specifically, when the agent observes $x_t$ from the environment, $\tau$ is sampled from the uniform distribution over $(0, 1)$, and it is subsequently fed into an online quantile network together with $x_t$. In this network, a $\tau$-embedding network $\nu(\cdot)$ transforms $\tau$ into embeddings, a LSTM feature extractor $\psi(\cdot)$ encodes the token sequence $x_t$ into a vector representation, and a fully-connected head $\phi(\cdot)$ produces the quantiles:

$$\widehat{\Theta}(x_t) = \phi(\psi(x_t) \odot \nu(\tau)) \in \mathbb{R}^{|\mathcal{A}| \times K},$$

where $\widehat{\Theta}(x_t)$ includes $\widehat{\theta}_k(x_t, a)$ for $a \in \mathcal{A}$ and $k = 1, \ldots, K$. With $\widehat{\Theta}(x_t)$ in hand, $\widehat{h}(x_t, a)$ can be computed via the QCM method for $a \in \mathcal{A}$. Then, the agent selects an exploratory action $a_t$ to enhance training efficiency:

$$a_t = \arg\max_{a \in \mathcal{A}} \left[ \widehat{Q}(x_t, a) + \lambda \sqrt{\widehat{h}(x_t, a)} \right], \tag{5}$$

where $\widehat{Q}(x, a)$ is computed by the online Q network of the DQN algorithm[6], and $\lambda$ is a tuning parameter to control the degree of risk-preference. See Figure 2 for a visual illustration. In this paper, the online Q network employs separate LSTM feature extractor and fully-connected head to transform the $x_t$ into Q values:

$$\widehat{Q}(x_t) = \phi(\psi(x_t)) \in \mathbb{R}^{|\mathcal{A}|},$$

where $\widehat{Q}(x_t)$ includes $\widehat{Q}(x_t, a)$ for $a \in \mathcal{A}$.

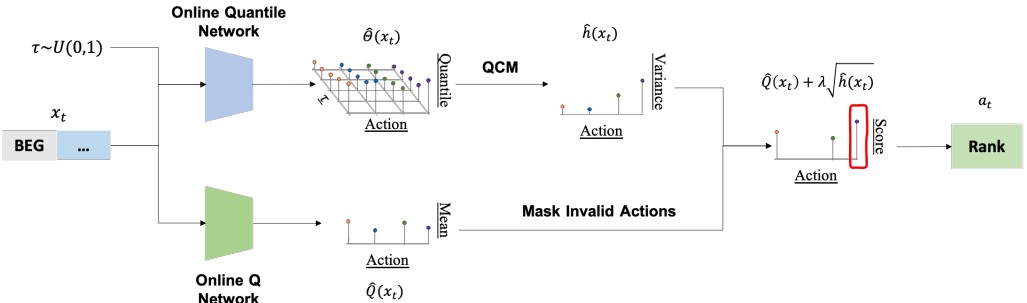

Figure 2: An illustration of action selection in our AlphaQCM framework

Motivated by (1), for each transition $(x_t, a_t, r_t, x_{t+1})$, the quantile temporal difference error (Dabney et al., 2018a) is defined as

$$\delta_{k,k',t} = r_t + \gamma \widetilde{\theta}_{k'} \left( x_{t+1}, \arg\max_{a' \in \mathcal{A}} \widetilde{Q}(x_t, a') \right) - \widehat{\theta}_k(x_t, a_t)$$

---

[6]Recall that, for a certain $x$, some actions are invalid, as mentioned in Section 3.2. Hence, we set the $\widehat{Q}(x, a) = -\infty$ for these invalid actions to mask them.

for $k' = 1, \ldots, K'$, where $\widetilde{Q}(x, a)$ is the target Q network output, and $\widetilde{\theta}_{k'}(x, a)$ is the target quantile network output. In the IQN framework, besides $\boldsymbol{\tau}$ for online quantile network, there is another $\widetilde{\boldsymbol{\tau}} = (\widetilde{\tau}_1, \ldots, \widetilde{\tau}_{K'})'$ independently sampled for the target quantile network. Notably, the target networks share the same architecture with their online counterparts but differ in network parameters, which are frozen but periodically synchronized from the online network.

Based on $\{\delta_{k,k',t}\}$, for a batch of $\{(x_t, a_t, r_t, x_{t+1})\}$, the Huber loss (Huber, 1964) for optimizing the online quantile network is defined as

$$\ell(\omega^*) = \sum_{t \in \text{batch}} \sum_{k=1}^{K} \sum_{k'=1}^{K'} \rho_{\tau_k}^{\kappa} \left( \delta_{k,k',t} \right),$$

where $\omega^*$ includes all network parameters in the online quantile network, and $\rho_{\tau}^{\kappa} \left( \delta_{k,k',t} \right) = |\tau - I\left( \delta_{k,k',t} < 0 \right)| \frac{L_{\kappa}\left( \delta_{k,k',t} \right)}{\kappa}$ with

$$L_{\kappa}\left( \delta_{k,k',t} \right) = \begin{cases} \frac{1}{2} \delta_{k,k',t}^2, & \text{if } |\delta_{k,k',t}| \leq \kappa \\ \kappa \left( |\delta_{k,k',t}| - \frac{1}{2}\kappa \right), & \text{otherwise} \end{cases},$$

$\kappa$ being a hyperparameter, and $I(\cdot)$ being the indicator function. Meanwhile, the online Q network is optimized by minimizing the sum of squared temporal difference errors:

$$\ell(\omega) = \sum_{t \in \text{batch}} \left[ r_t + \gamma \max_{a' \in \mathcal{A}} \widetilde{Q}(x_{t+1}, a') - \widehat{Q}(x_t, a_t) \right]^2,$$

where $\omega$ represents all network parameters in the online Q network.

Note that we equip the agent with the prioritized experience replay method (Schaul et al., 2016). While the classical experience replay method samples transitions uniformly from a replay memory, the prioritized experience replay method improves sampling efficiency by replaying more frequently transitions from which there is more to learn. Specifically, it samples transitions with prior probability $p_t$ related to the last encountered quantile temporal difference error, where $p_t \propto \left| \sum_k \sum_{k'} \rho_{\tau_k}^{\kappa} \left( \delta_{k,k',t} \right) \right|^{\eta}$ with $\eta$ being a hyperparameter. Clearly, the agent equipped with prioritized experience replay can utilize the transitions guided by the QCM method more efficiently.

To save space, the hyperparameters used in our AlphaQCM method are specified in Appendix F, while some unmentioned technical details are consistent with our backbone.

## 4 EXPERIMENTS

### 4.1 DATASET, COMPARISON METHODS AND EVALUATION METRIC

Given that Yu et al. (2023) is the most closely related work, our experiments are also conducted on Chinese A-share stock market datasets to capture the 20-day future stock returns. To evaluate the impact of the complexity of the considered financial system on performance, we consider three different stock pools: (1) the largest 300 stocks (CSI300), (2) the largest 500 stocks (CSI500), and (3) all stocks (Market). As one might expect, the more stocks involved in the dataset, the more challenging it becomes to discover synergistic formulaic alphas, as the system becomes more complex and chaotic. Moreover, each dataset is split chronologically into a training set (2010/01/01 to 2019/12/31), a validation set (2020/01/01 to 2020/12/31), and a test set (2021/01/01 to 2022/12/31).

We consider the following four kinds of baseline methods for comparison:

1. **Alpha101** (human-designed formulaic alphas): Fix the alpha pool as the formulaic alphas provided by (Kakushadze, 2016), and fit a linear model to form a mega-alpha.

2. **MLP**, **XGBoost**, **LightGBM** (ML-based non-formulaic alphas): Use the MLP model, XGBoost model, or LightGBM model to form a mega-alpha.

3. **GP w/o filter**, **GP w/ filter** (GP-based formulaic alphas): Use the GP method to generate expressions and apply top-$P$ performing alphas without or with a mutual IC filter to form a mega-alpha.

4. **AlphaGen** (RL-based formulaic alphas): Use the AlphaGen method to find the optimal alpha pool and then form a linear mega-alpha.

To account for the effect of stochasticity in the training process, we evaluate each indeterministic experimental combination with 10 different random seeds. More details and the rationale for choosing these baseline methods are provided in Appendix G.

Following the existing literature Yu et al. (2023), Cui et al. (2021), and Lin et al. (2019a;b), we choose the IC as the most important metric to evaluate the out-of-sample performance.

### 4.2 IMPACT OF METHODS

We first assess how the alpha-generating methods affect the out-of-sample performance of the formed meta-alphas. For a fair comparison, we regard $P$ (alpha pool size) as a hyperparameter and choose it based on performance on the validation set. Table 1 reports the means and standard deviations of IC values across eight different methods in CSI300, CSI500, and Market datasets. From this table, we can draw the following conclusions:

(1) Our AlphaQCM method with the highest IC value outperforms all competitors, regardless of the stock pool considered. Moreover, the AlphaGen method, which is the most closely related baseline method, ranks second. The advantage of the AlphaQCM method over the AlphaGen method becomes more significant as the number of stocks in the dataset increases. This superior performance may be attributed to the fact that the AlphaGen method totally ignores the non-stationary and reward-sparse issues, while these issues become more pronounced as the the concerned system becomes more complex and chaotic.

(2) For the GP method, incorporating the mutual IC filter improves its performance in the CSI300 and CSI500 datasets but results in worse performance in the Market dataset. This finding highlights the limitation of using the mutual IC filter to find synergistic formulaic alphas when a large number of stocks are involved, as commonly observed in modern portfolio selection.

(3) The machine learning methods slightly underperform the Alpha101 method in the CSI300 and CSI500 datasets, while they outperform the Alpha101 method in the Market dataset, with the exception of the MLP method. This observation suggests that when facing big data, the formulaic alphas discovered by human experts may lose their advantage over the non-formulaic alphas. However, the RL-based methods surpass both human experts and machine learning methods, with the discovered alphas remaining interpretable.

Table 1: Out-of-sample IC values across different methods.

| Method | CSI300 Mean | CSI300 Std | CSI500 Mean | CSI500 Std | Market Mean | Market Std |
|---|---|---|---|---|---|---|
| Alpha101 | 3.44% | - | 4.38% | - | 3.15% | - |
| MLP | 1.99% | 0.24% | 2.72% | 0.65% | 2.81% | 0.72% |
| XGBoost | 3.19% | 0.81% | 4.31% | 0.96% | 4.07% | 1.22% |
| LightGBM | 2.93% | 0.76% | 4.16% | 0.81% | 4.28% | 0.93% |
| GP w/o filter | 2.01% | 1.46% | 1.79% | 1.62% | 1.32% | 2.01% |
| GP w/ filter | 3.71% | 2.01% | 4.52% | 1.93% | 0.84% | 2.27% |
| AlphaGen | 8.13% | 0.94% | 8.08% | 1.23% | 6.04% | 1.78% |
| AlphaQCM | **8.49%** | **1.03%** | **9.55%** | **1.16%** | **9.16%** | **1.61%** |

### 4.3 IMPACT OF QCM METHOD AND DRL BACKBONES

One of our key contributions is using the QCM method to overcome the challenges of non-stationarity and reward-sparsity. To specify the contribution of the QCM method to empirical performance, we consider the following two competitors:

1. **No variance**: Fix $\lambda = 0$ in (5) to remove the impact of variance in action selection, meaning that the QCM method has no effect in the training process.

2. **Vanilla variance**: Replace $\widehat{h}(x, a)$ in (5) with the vanilla quantile-based variance estimator.

Moreover, since our AlphaQCM method relies on the DRL backbone employed to learn quantiles, a natural question arises: which type of DRL backbone is more effective for alpha mining? To answer this question, we alter the AlphaQCM method by choosing the QRDQN algorithm (Dabney et al., 2018b) as the backbone, which serves as another benchmark.

Table 2 reports the results of the conducted ablation study. From this table, we observe that regardless of the backbone employed, using the QCM variance always earns the highest IC value, whereas the action selection based on the vanilla variance results in better performance than that based on no variance only in CSI500 and Market datasets. Additionally, from the viewpoint of convergence performance, using variance to encourage exploration consistently brings lower standard deviations of IC values. These findings imply that using variance for exploration enhances training efficiency and convergence performance in the alpha discovery MDP, and at the same time, a better quality of variance estimation leads to a better empirical performance.

We also find that the IQN algorithm outperforms the QRDQN algorithm in most cases, except for no variance in Market dataset and vanilla variance in CSI500 dataset. However, the differences in performance between the IQN and QRDQN backbones are marginal in these two exceptions. These empirical results demonstrate the superior performance of using the IQN algorithm as the backbone for the AlphaQCM method compared to the QRDQN algorithm.

Table 2: Out-of-sample IC values across different action selection criteria and DRL backbones.

| Variance | CSI300 | | CSI500 | | Market | |
|---|---|---|---|---|---|---|
| | Mean | Std | Mean | Std | Mean | Std |
| Panel A: QRDQN as backbone | | | | | | |
| No | 6.96% | 1.64% | 8.54% | 1.56% | 7.06% | 1.92% |
| Vanilla | 6.14% | 1.23% | 8.80% | 1.34% | 7.60% | 1.17% |
| QCM | 7.59% | 0.81% | 9.08% | 1.07% | 9.12% | 1.74% |
| Panel B: IQN as backbone | | | | | | |
| No | 7.17% | 2.40% | 8.58% | 1.47% | 7.04% | 1.82% |
| Vanilla | 6.16% | 1.73% | 8.75% | 1.03% | 8.42% | 1.59% |
| QCM | **8.49%** | **1.03%** | **9.55%** | **1.16%** | **9.16%** | **1.61%** |

## 5 CONCLUSION

This paper proposes a novel DRL method, AlphaQCM, for alpha discovery in the realm of big market data. Unlike the existing methods in the literature, the key idea of the AlphaQCM method relies on the unbiased estimation of variance derived from potentially biased quantiles. This approach enables the efficient alpha discovery in the non-stationary and reward-sparse MDP. To implement the AlphaQCM method, we employ the IQN algorithm as the backbone to obtain quantiles, while approximating the Q function using the DQN algorithm. Through extensive experiments on three real-world datasets, we demonstrate the superior advantage of our AlphaQCM method over the previous state-of-the-art alpha discovery approaches. The ablation studies further highlight the contribution of the QCM method, the robustness of DRL backbone, the independence from human domain knowledge and the robustness of parameter size[7].

Overall, the AlphaQCM method serves as a powerful tool for discovering synergistic formulaic alphas, with its superior capability allowing for non-stationarity and reward-sparsity in the alpha discovery process.

---

[7]See Appendixes H and I for these ablation studies.

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

## A  EVALUATION METRIC FOR ALPHAS

To measure the performance of an alpha, it is a common practice to use the IC as the evaluation metric. For a given alpha pool $\mathcal{F}$, the IC for this alpha pool $\mathcal{F}$ is defined as:

$$\mathrm{IC}(\mathcal{F}) = \mathrm{Mean}[\mathrm{Corr}(\widehat{\boldsymbol{\alpha}}_s, \boldsymbol{y}_s)]$$

$$\equiv \frac{1}{S} \sum_{s=1}^{S} \frac{\sum_{i=1}^{N} (\widehat{\alpha}_{i,s} - \bar{\alpha}_s)(y_{i,s} - \bar{y}_s)}{\sqrt{\sum_{i=1}^{N} (\widehat{\alpha}_{i,s} - \bar{\alpha}_s)^2 \sum_{i=1}^{N} (y_{i,s} - \bar{y}_s)^2}}$$

where $\{(\widehat{\boldsymbol{\alpha}}_s, \boldsymbol{y}_s) : s = 1, \ldots, S\}$ is the sample of meta-alphas based on $\mathcal{F}$ and true returns $\{\boldsymbol{y}_s\}$, $\widehat{\alpha}_{i,s}$ and $y_{i,s}$ are $i$-th element of $\widehat{\boldsymbol{\alpha}}_s$ and $\boldsymbol{y}_s$, respectively,

$$\bar{\alpha}_s = \frac{1}{N} \sum_{i=1}^{N} \widehat{\alpha}_{i,s} \quad \text{and} \quad \bar{y}_s = \frac{1}{N} \sum_{i=1}^{N} y_{i,s}.$$

Here, $\mathrm{Corr}(\widehat{\boldsymbol{\alpha}}_s, \boldsymbol{y}_s)$ calculates the cross-sectional correlation to measure the predictive power of $\mathcal{F}$ at time $s$, while the $\mathrm{Mean}$ operator is used to obtain a time-averaged result over the period containing all of $S$ timepoints.

## B  AVAILABLE FEATURES AND OPERATORS

Table B.3: Description of available features and operators.

| Tokens | Description |
|---|---|
| Features | |
| Open/High/Low/Close/Vwap /Volume | Opening/high/low/closing/vwap price or volume of stock $i$ at time $s$. |
| Constant | Number from $\{-30, -10, -5, -2, -1, -0.5, -0.01, 0.01, 0.5, 1, 2, 5, 10, 30\}$. |
| Time delta | Integer from $\{10, 20, 30, 40, 50\}$, which is always used in the time-series operators. |
| Time-series Operators | |
| $Ref(u_{i,s}, d)$ | Return the value of $u_{i,s-d}$, where $d$ is the time delta and $u_{i,s}$ is a feature of stock $i$ at time $s$. |
| $TsRank(u_{i,s}, d)$ | Return the rank of $u_{i,s}$ among $\{u_{i,s}, \ldots, u_{i,s-d}\}$ |
| $Mean/Med/Sum/Std/Var(u_{i,s}, d)$ | Return the mean/median/sum/standard deviation/variance of $\{u_{i,s}, \ldots, u_{i,s-d}\}$. |
| $Max/Min(u_{i,s}, d)$ | Return the maximum/minimum value of $\{u_{i,s}, \ldots, u_{i,s-d}\}$. |
| $WMA/EMA(u_{i,s}, d)$ | Return the weighted/exponentially weighted moving average of $\{u_{i,s}, \ldots, u_{i,s-d}\}$. |
| $Cov/Corr(u_{i,s}, z_{i,s}, d)$ | Return the covariance/correlation between $u_{i,s}$ and $z_{i,s}$ based on samples $\{(u_{i,s}, z_{i,s}) \ldots, (u_{i,s-d}, z_{i,s-d})\}$, where $u_{i,s}$ and $z_{i,s}$ are features of stock $i$ at time $s$. |
| Cross-sectional Operators | |
| $Sign(u_{i,s})$ | Return 1 if the value of $u_{i,s}$ is positive, otherwise return 0. |
| $Abs(u_{i,s})$ | Return the absolute value of $u_{i,s}$. |
| $Log(u_{i,s})$ | Return the logarithmic value of $u_{i,s}$. |
| $Rank(u_{i,s})$ | Return the rank of $u_{i,s}$ among $\{u_{1,s}, \ldots, u_{N,s}\}$. |
| $Add/Sub/Mul/Div(u_{i,s}, z_{i,s})$ | Return the the result of adding/subtracting/multiplying /dividing $u_{i,s}$ and $z_{i,s}$. |
| $Greater/Less(u_{i,s}, z_{i,s})$ | Return the greater/less value of $u_{i,s}$ and $z_{i,s}$. |

## C ALGORITHM FOR CALCULATING REWARD

Algorithm C.1 is the pseduo code for calculating reward $r_t$.

---

**Algorithm C.1:** Pseduo code for calculating $r_t$.

---

**Input:** training samples $\{(\boldsymbol{H}_{s-1}, \boldsymbol{y}_s)\}$, new alpha $f_{\text{new}}$, and alpha set $\mathcal{F} = \{f_1, f_2, \ldots, f_{P^*}\}$;
**Output:** updated alpha set $\mathcal{F}^*$ and reward $r_t$;
1 # Calculate alpha values and normalize them;
2 $f_{P^*+1} \leftarrow f_{\text{new}}$;
3 $\boldsymbol{\alpha}_{p,s} \leftarrow f_p(\boldsymbol{H}_{s-1})$ for $p = 1, \ldots, P^* + 1$;
4 $\boldsymbol{\alpha}_{p,s} \leftarrow \frac{\boldsymbol{\alpha}_{p,s} - \text{Mean}(\boldsymbol{\alpha}_{p,s})}{\text{Std}(\boldsymbol{\alpha}_{p,s})}$ for $p = 1, \ldots, P^* + 1$;
5 # Fit a linear model based on the extended alpha pool;
6 $\boldsymbol{A}_s \leftarrow [\boldsymbol{\alpha}_{1,s}, \ldots, \boldsymbol{\alpha}_{P^*+1,s}] \in \mathbb{R}^{N \times (P^*+1)}$;
7 $\widehat{\boldsymbol{\beta}} \leftarrow \arg\min_{\boldsymbol{\beta}} \sum_s \|\boldsymbol{y}_s - \boldsymbol{A}_s\boldsymbol{\beta}\|^2 \in \mathbb{R}^{P^*+1}$ ;
8 # Obtain the updated alpha pool and meta-alpha;
9 **if** $P^* + 1 \leq P$ **then**
10     $\mathcal{F}^* \leftarrow \{f_1, \ldots, f_{P^*+1}\}$;
11     $\widehat{\boldsymbol{\alpha}}_s \leftarrow \boldsymbol{A}_s\widehat{\boldsymbol{\beta}} \in \mathbb{R}^N$;
12 **else**
13     $\bar{p} \leftarrow \arg\min_p |\widehat{\beta}_p|$, where $\widehat{\boldsymbol{\beta}} = (\widehat{\beta}_1, \ldots, \widehat{\beta}_{P^*+1})'$;
14     $\mathcal{F}^* \leftarrow \{f_1, \ldots, f_{\bar{p}-1}, f_{\bar{p}+1}, \ldots, f_{P^*+1}\}$;
15     $\widehat{\boldsymbol{\alpha}}_s \leftarrow \boldsymbol{A}_s\widehat{\boldsymbol{\beta}} - \boldsymbol{\alpha}_{\bar{p},s}\widehat{\beta}_{\bar{p}} \in \mathbb{R}^N$;
16 # Calculate IC based on $\mathcal{F}^*$ and reward $r_t$;
17 $\text{IC}(\mathcal{F}^*) \leftarrow \text{Mean}(\text{Corr}(\widehat{\boldsymbol{\alpha}}_s, \boldsymbol{y}_s))$;
18 $r_t \leftarrow \text{IC}(\mathcal{F}^*) - \text{IC}(\mathcal{F})$;
19 **return** $\mathcal{F}^*$ and $r_t$.

---

## D ASYMPTOTIC PROPERTIES

To establish the asymptotic properties of the QCM estimators, some mild assumptions are required. For ease of presentation, define

$$
\boldsymbol{Z} = \begin{pmatrix}
1 & z_1 & z_1^2 - 1 & z_1^3 - 3z_1 \\
1 & z_2 & z_2^2 - 1 & z_2^3 - 3z_2 \\
\vdots & \vdots & \vdots & \vdots \\
1 & z_K & z_K^2 - 1 & z_K^3 - 3z_K
\end{pmatrix},
$$

$$
\boldsymbol{\beta}(x, a) = \begin{pmatrix}
\zeta(x,a) + Q^*(x,a) \\
\sqrt{h(x,a)} \\
\frac{\sqrt{h(x,a)s(x,a)}}{6} \\
\frac{\sqrt{h(x,a)[k(x,a)-3]}}{24}
\end{pmatrix}, \text{ and } \boldsymbol{\varepsilon}(x,a) = \begin{pmatrix}
\varepsilon_1(x,a) \\
\varepsilon_2(x,a) \\
\vdots \\
\varepsilon_K(x,a)
\end{pmatrix},
$$

where some notations are given in Section 3.3.1. Then, Proposition 3.1 holds if the following two classical assumptions in the regression literature hold:

**Assumption 1.** $\boldsymbol{Z}'\boldsymbol{Z}$ *is positive definite.*

**Assumption 2.** $\boldsymbol{Z}'\boldsymbol{\varepsilon}(x,a)/K \xrightarrow{p} \boldsymbol{0}$ *as* $K \to \infty$.

Moreover, if we assume $\boldsymbol{Z}'\boldsymbol{\varepsilon}(x,a)/K \longrightarrow \boldsymbol{0}$ almost surely as $K \to \infty$ in Assumption 2, all of convergence results in Proposition 3.1 hold almost surely. Interested readers could refer to Zhang & Zhu (2023) for more comprehensive analysis on the QCM method.

# E  VANILLA QUANTILE-BASED VARIANCE ESTIMATOR

Recall that the widely used quantile representation $Z_{\boldsymbol{\theta},\boldsymbol{\tau}}(x,a)$, defined in (2), approximates $Z^*(x,a)$ with a mixture of Dirac distributions. With this approximation, $\mathrm{Var}[Z^*(x,a)]$ can be approximated by the vanilla quantile-based variance estimator $\mathrm{Var}[Z_{\widehat{\boldsymbol{\theta}},\boldsymbol{\tau}}(x,a)]$, where

$$\mathrm{Var}[Z_{\widehat{\boldsymbol{\theta}},\boldsymbol{\tau}}(x,a)] = \sum_{k=1}^{K}(\tau_k - \tau_{k-1})\left\{\widehat{\theta}_k(x,a) - \mathbb{E}[Z_{\widehat{\boldsymbol{\theta}},\boldsymbol{\tau}}(x,a)]\right\}^2.$$

Here, $\widehat{\theta}_k(x,a)$ for $k=1,\ldots,K$ are the $\tau_k^*$-th quantile estimates, and

$$\mathbb{E}[Z_{\widehat{\boldsymbol{\theta}},\boldsymbol{\tau}}(x,a)] = \sum_{k=0}^{K-1}(\tau_{k+1} - \tau_k)\widehat{\theta}_k(x,a).$$

Intuitively, $\mathbb{E}[Z_{\widehat{\boldsymbol{\theta}},\boldsymbol{\tau}}(x,a)]$ can be called the vanilla quantile-based Q estimator.

# F  HYPERPARAMETERS

## F.1  NETWORK

Our network-related hyperparameters are consistent with those in Yu et al. (2023) for a fair comparison. Specifically, both in the online Q network and online quantile network, the LSTM feature extractor $\psi(\cdot)$ has a 2-layer structure with a hidden layer dimension of $128$ with dropout rate of $0.1$, and the fully connected heads have two hidden layers of $64$ dimensions. Moreover, the $\tau$-embedding network maps each quantile level into a $64$-dimensional embedding, as defined in Dabney et al. (2018a).

## F.2  DRL

Besides the network-related hyperparameters, some additional hyperparameters that our DRL algorithm inherits from the IQN and DQN algorithms are listed in Table F.4.

Table F.4: Additional hyperparameters.

| Hyperparameter | Values |
|---|---|
| Min history to start learning | 10,000 |
| $\epsilon$-greedy | 0.01 |
| Memory size | 100,000 |
| Learning rate | 5e-5 |
| Optimizer | Adam |
| Online network update interval (replay period) | 1 |
| Target network update interval | 5,000 |
| Batch size | 128 |
| $K$ (length of $\boldsymbol{\tau}$) | 64 |
| $K'$ (length of $\tilde{\boldsymbol{\tau}}$) | 64 |
| $\kappa$ (constant in the Huber loss) | 1.0 |
| $\eta$ (constant in the prior probability) | 0.5 |
| $\lambda$ (tuning parameter) | 0.5, 1, 2 |
| $P$ (Alpha pool size) | 10, 20, 50, 100 |
| Total step | 250,000 ($P = 10$), 300,000 ($P = 20$), 350,000 ($P = 50$), 400,000 ($P = 100$) |

## G   BASELINE METHODS

To evaluate the performance of our AlphaQCM algorithm, we consider four types of baseline methods in Section 4.1. The first type of baseline methods aims to measure human-level performance in alpha discovery, serving as a benchmark for the remaining machine-based methods. In this category, we apply the well-known Alpha101 method as a representative, which employs the 101 formulaic alphas from Kakushadze (2016) to construct mega-alphas. To ensure model interpretable, these alphas are combined linearly based on the samples in the training and validation sets[8] using a squared loss function. The comparison between the AlphaQCM and Alpha101 methods aims to evaluate whether the RL-based alpha generator can outperform human experts.

Next, the second type of baseline methods directly applies some end-to-end machine learning models to capture the linkage between stock features and their future returns. Clearly, these machine-learning-based alphas lack interpretability but are straightforward to generate. Following Yu et al. (2023), we implement the MLP, XGBoost, and LightGBM methods using the open-source library Qlib (Yang et al., 2020), with pre-specified hyperparameters. The comparison between the AlphaQCM method and the second type of baseline methods is to verify whether the combination of interpretable alphas can outperform complex and non-interpretable alphas.

Moreover, the third type of baseline methods depends on the genetic programming algorithm. This type of methods employs the GP method to generate alphas in a one-by-one manner, with the IC being the fitness measure. Specifically, for the GP w/o filter method, the top-$P$ generated alphas with the highest ICs within the training set are used to form a mega-alpha via a linear model, which is fitted on the validation set. In contrast, the GP w/ filter method selects the top-$P$ performing alphas with an additional mutual-IC filter, ensuring that any pair of alphas in the set does not have a mutual IC higher than $0.7$. The comparison between the AlphaQCM and the GP-based methods aims to probe the limitations of the GP method when involving large populations.

Lastly, the AlphaGen method (Yu et al., 2023) is the most closely related competitor to our AlphaQCM method, which also belongs to the category of RL-based methods. The primary difference between these two methods is the algorithm used for discovering alphas. The comparison between the AlphaQCM and AlphaGen methods aims to check whether the simple PPO algorithm is adequate for such a non-stationary and reward-sparse MDP.

## H   ADDITIONAL EXPERIMENT: IMPACT OF DOMAIN KNOWLEDGE

Until now, the AlphaQCM method has discovered formulaic alphas in a completely data-driven manner, neglecting the valuable insights offered by economic experts in the field of alpha discovery. One potential approach to incorporate the domain knowledge of experts into our AlphaQCM algorithm is to encode the formulaic alphas proposed in Kakushadze (2016) into token sequences and initialize the replay memory with these corresponding trajectories[9]. In other words, we restrict the agent to first learn from the alphas discovered by human experts and then find new alphas in a data-driven way.

To check whether the domain knowledge enhances the efficiency of the AlphaQCM method, we report the performance of AlphaQCM method with and without domain knowledge in Table H.5. From this table, we observe that there is some initial gain when the AlphaQCM method leverages domain knowledge, as the AlphaQCM method with domain knowledge outperforms the one without domain knowledge in Panels A and B. However, with more agent-environment interactions and training as in Panels C and D, the AlphaQCM method in a completely data-driven manner achieves higher IC values. This is perhaps because the agent with domain knowledge is prone to fall into local optima by imitating the experts. Hence, to achieve better final performance, we suggest applying the AlphaQCM method without domain knowledge.

---

[8]We download the data for 101 formulaic alphas from "www.ricequant.com".

[9]Specifically, as in Figure 1, we encode the mathematical expression of "Alpha#4" into its RPN representation, which is a token sequence basically. Then, the generating process of this token sequence is regarded as a trajectory for agent-environment interactions, which are used to initiate the replay memory. Unfortunately, the formulation of 101 formulaic alphas requires some sophisticated operators and features, which are not considered in this paper. We discard those formulaic alphas including unconsidered features and operators.

Table H.5: Out-of-sample IC values of the AlphaQCM method with or without domain knowledge.

| Domain | CSI300 | | CSI500 | | Market | |
|---|---|---|---|---|---|---|
| | Mean | Std | Mean | Std | Mean | Std |
| Panel A: After 10% Training | | | | | | |
| w/ | 4.93% | 0.71% | 5.76% | 0.68% | 6.02% | 0.92% |
| w/o | 4.27% | 1.75% | 5.68% | 1.51% | 5.87% | 1.34% |
| Panel B: After 20% Training | | | | | | |
| w/ | 6.32% | 1.29% | 7.01% | 1.77% | 6.85% | 1.44% |
| w/o | 5.54% | 0.78% | 6.43% | 1.38% | 6.43% | 2.83% |
| Panel C: After 50% Training | | | | | | |
| w/ | 6.41% | 1.47% | 7.15% | 1.22% | 7.33% | 1.56% |
| w/o | 6.82% | 1.35% | 7.57% | 2.12% | 7.48% | 1.84% |
| Panel D: After 100% Training | | | | | | |
| w/ | 8.17% | 1.17% | 8.96% | 1.51% | 8.60% | 1.23% |
| w/o | **8.49%** | **1.03%** | **9.55%** | **1.16%** | **9.16%** | **1.61%** |

# I  ADDITIONAL EXPERIMENT: IMPACT OF PARAMETER SIZE

As mentioned in Appendix F.1, the network-related hyperparameters for both the AlphaGen and AlphaQCM methods are kept consistent to ensure a fair comparison. However, due to differences in their network architectures, the total parameter size of these two methods varies. Specifically, the AlphaGen method employs two types of networks: an actor network and a critic network, with a total of $298,609$ trainable parameters. In contrast, the AlphaQCM method utilizes four networks: an online Q-network, an online quantile network, and their respective target networks (with frozen parameters), resulting in $572,000$ trainable parameters. Thus, the AlphaQCM method has nearly twice the number of trainable parameters compared to AlphaGen, with the same network-related hyperparameters.

To examine whether the superior performance of the AlphaQCM method over the AlphaGen method stems from its ability to handle non-stationary and reward-sparse alpha discovery MDP rather than the increased model complexity, we designed two additional variants: a larger AlphaGen method and a smaller AlphaQCM method. The larger AlphaGen method has $564,953$ trainable parameters, achieved by setting the hidden dimensions of the LSTM feature extractor to 180, with fully connected heads comprising two hidden layers of 90 dimensions each. Conversely, the smaller AlphaQCM method has $297,070$ trainable parameters, achieved by setting the hidden dimensions of the LSTM network to 90, with fully connected heads comprising two hidden layers of 48 dimensions each. These configurations ensure that the larger AlphaGen method aligns with the parameter size of the proposed AlphaQCM method, while the smaller AlphaQCM method matches the complexity of the original AlphaGen method.

Table I.6: Out-of-sample IC values for methods with different parameter sizes.

| Method | CSI300 | | CSI500 | | Market | |
|---|---|---|---|---|---|---|
| | Mean | Std | Mean | Std | Mean | Std |
| Panel A: Small-parameter method (nearly 300K parameters) | | | | | | |
| AlphaGen | 8.13% | 0.94% | 8.08% | 1.23% | 6.04% | 1.78% |
| AlphaQCM | 8.23% | 0.90% | **9.68%** | **1.07%** | 9.12% | 1.47% |
| Panel B: Large-parameter method (nearly 570K parameters) | | | | | | |
| AlphaGen | 8.09% | 1.33% | 8.40% | 1.48% | 6.67% | 2.03% |
| AlphaQCM | **8.49%** | **1.03%** | 9.55% | 1.16% | **9.16%** | **1.61%** |

Table I.6 reports the IC values for methods with different parameter sizes. From this table, we first observe that both methods are robust to changes in parameter size. Additionally, the AlphaQCM method consistently outperforms the AlphaGen method across different parameter scales, with substantial performance gaps. These findings further emphasize the capability of the AlphaQCM method in solving the non-stationary and reward-sparse alpha discovery MDP effectively.

