# OpenReview forum: "AlphaQCM: Alpha Discovery with Distributional Reinforcement Learning"
_ICLR.cc/2025/Conference — Submitted to ICLR 2025_

### Official Review · Reviewer_JqfW · 2024-10-31

**Soundness:** 2
**Presentation:** 2
**Contribution:** 1
**Rating:** 3
**Confidence:** 4

**Summary:**

The paper presents a new approach to "alpha discovery" in finance using deep reinforcement learning. Alpha discovery is the problem of finding a function that maps stock histories onto predictive signals for future returns. The authors claim that this problem is challenging because the domain is non stationary and reward stationarity. The authors propose a distributional reinforcement learning approach called AlphaQCM. Some theoretical analysis is provided and experiments compare the proposed method with some baselines.

**Strengths:**

- The writing is generally clear.
- The problems of nonstationarity and reward sparsity are interesting.

**Weaknesses:**

The most salient weakness is that the paper appears to be written for a finance audience. Please see questions below.

**Questions:**

Questions:
- What is a synergistic formulaic alpha?
- What is alpha mining?
- It is unclear whether the paper aims to contribute to financial modeling or reinforcement learning.
- Why is the definition of "alpha" in a footnote. For a technical conference, given that this is the goal of the work, I would expect the authors to explain what this is and why it is important first.
- What does it mean for alpha to be subtle and intricate?
- Line 035, I don't know that "surpass" conveys the intended meaning here.
- What is AlphaGen?
- Neither nonstationarity nor reward sparsity are adequately defined in the introduction.
- It is not clear why these methods are chosen for comparison. I see that we are supposed to look in an Appendix for the justification, but why wouldn't this be in the main text?
- Specifically, I would expect to see, for a machine learning audience, a justification for the alternative models in terms of their prior performance in non stationary and/or sparse reward settings.
- Why are the only datasets financial in nature? There are other examples of non stationary sparse domains. I would expect those to be included, if the goal of the algorithm is to overcome those challenges.

---

> ### Author Response · Authors · 2024-11-20
> **Response for Reviewer JqfW (1/2)**
>
> Thank you for your comments. To improve the quality of this paper, we revised the manuscript according to your comments. Below is our point-by-point response to your questions:
>
> **Q1: What is a synergistic formulaic alpha?**
>
> A: An important question. As outlined in Lines 042--045 of the revised manuscript (Lines 040--043 of the original manuscript), a synergistic formulaic alpha is defined from two perspectives:
> 1. *Formulaic*: a formulaic alpha can be represented by a certain simple formula;
> 2. *Synergistic*: a synergistic alpha can be combined with other synergistic alphas into an effective meta-alpha via some interpretable models (e.g., linear models in Section 3).
>
> Our goal is to identify several formulaic alphas that work synergistically, enabling us to construct a linear, predictive meta-alpha. Similar approaches can be found in [1], [2] and [3].
>
> Hopefully, the definition of a synergistic formulaic alpha is clear to you now, and you are satisfied with our efforts.
>
> **Q2: What is alpha mining?**
>
> A: Great question. Essentially, alpha mining is closely related to alpha discovery, implying that effective formulaic alphas are hidden like ore beneath the surface, requiring extraction through careful analysis. A similar metaphor is used in [1].
>
> To enhance the clarity of this paper, we have replaced all instances of "alpha mining" with "alpha discovery" in the revised manuscript.
>
> Hopefully, you are satisfied with these changes.
>
> **Q3: It is unclear whether the paper aims to contribute to financial modeling or reinforcement learning.**
>
> A: This paper makes contributions to both fields. Primarily, it addresses the non-stationarity and reward sparsity in the alpha discovery problem by developing a novel reinforcement learning algorithm. Furthermore, this algorithm has broader applicability and can be extended to other non-stationary or reward-sparse MDPs.
>
> To clarify and emphasize these contributions, we have revised the manuscript accordingly (Lines 059–-061 of the revised manuscript).
>
> Hopefully, the contribution of this paper is clear to you now, and you are satisfied with our efforts.
>
> **Q4: Why is the definition of ``alpha'' in a footnote. For a technical conference, given that this is the goal of the work, I would expect the authors to explain what this is and why it is important first.**
>
> A: Thank you for raising this important point. Since "alpha" is a commonly used term in related literature, we initially included its definition in a footnote for brevity in the original manuscript.
>
> In the revised manuscript, we have highlighted its definition and significance more prominently to underscore its central role in our work (Lines 031–-032 of the revised manuscript).
>
> Hopefully, you are satisfied with these changes.
>
> **Q5: What does it mean for alpha to be subtle and intricate?**
>
> A: Due to the influence of investor behavior and psychology, some effective alphas may take on complex functional forms, capturing the nuanced and intricate relationships inherent in financial markets. These alphas are difficult to formalize using traditional empirical financial methods, which is why we refer to them as "subtle and intricate".
>
> We have emphasized this explanation in Lines 033–034 of the revised manuscript.
>
> Hopefully, you are satisfied with these changes.
>
> **Q6: Line 035, I don't know that ``surpass'' conveys the intended meaning here.**
>
> A: In the corresponding paragraph, our intention is to highlight that a well-trained machine learning model functions as a complex alpha (i.e., a function mapping noisy historical stock data into predictive signals for future returns) and significantly outperforms classical financial models.
>
> We have added more descriptive context in the revised manuscript (Lines 037–-038) to clarify this point.
>
> Hopefully, you are satisfied with our efforts.
>
> **Q7: What is AlphaGen?**
>
> A: As mentioned in Lines 049--050 and 93 of the revised manuscript (Lines 047 and 110 of the original manuscript), the AlphaGen method is a reinforcement learning method for alpha discovery proposed by [1]. While it achieves state-of-the-art performance in alpha discovery, it does not address non-stationarity or reward sparsity, leaving significant room for improvement. In our study, the AlphaGen method serves as the closest baseline method for comparison.

---

> > ### Author Response · Authors · 2024-11-20
> > **Response for Reviewer JqfW (2/2)**
> >
> > **Q8: Neither nonstationarity nor reward sparsity are adequately defined in the introduction.**
> >
> > A: Thanks for your good comment. To provide better clarity upfront, we add the definitions of these concepts in Lines 053--055 and 058 in the revised manuscript, while more detailed discussions are provided in Section 3.2.
> >
> > Hopefully, you are satisfied with these changes.
> >
> > **Q9: It is not clear why these methods are chosen for comparison. I see that we are supposed to look in an Appendix for the justification, but why wouldn't this be in the main text?**
> >
> > A: Thank you for raising this point. In this paper, the comparison methods are categorized into four groups: human-designed formulaic alphas, machine-learning-based non-formulaic alphas, genetic-programming-based formulaic alphas, and RL-based formulaic alphas.
> >
> > To enhance clarity and accessibility, we have added a brief explanation of the rationale for selecting these methods in the main text of the revised manuscript (Lines 425–434).
> >
> > Hopefully, you are satisfied with our efforts.
> >
> > **Q10: Specifically, I would expect to see, for a machine learning audience, a justification for the alternative models in terms of their prior performance in non stationary and/or sparse reward settings.**
> >
> > A: Great question. Here, the AlphaGen method employs an alternative reinforcement learning algorithm (PPO) in this non-stationary and reward-sparse MDP. As demonstrated in Table 1, our AlphaQCM method demonstrates superior performance, effectively addressing these challenges and highlighting the contributions of this work.
> >
> > **Q11: Why are the only datasets financial in nature? There are other examples of non stationary sparse domains. I would expect those to be included, if the goal of the algorithm is to overcome those challenges.**
> >
> > A: While there are indeed many non-stationary and reward-sparse MDPs beyond the financial domain, our work is specifically motivated by challenges in the alpha discovery problem, where previous methods have struggled. We focused on financial datasets for this reason. However, we welcome any suggestions for additional tasks where you would like us to test our algorithm, as this could provide valuable insights into its broader applicability.
> >
> > **References:**
> >
> > [1] Shuo Yu, Hongyan Xue, Xiang Ao, Feiyang Pan, Jia He, Dandan Tu, and Qing He. Generating synergistic formulaic alpha collections via reinforcement learning. In Proceedings of the 29th ACM SIGKDD Conference on Knowledge Discovery and Data Mining, KDD ’23, pp. 5476–5486. Association for Computing Machinery, 2023.
> >
> > [2] Xiaoming Lin, Ye Chen, Ziyu Li, and Kang He. Stock alpha mining based on genetic algorithm. Technical report, Huatai Securities Research Center, 2019a.
> >
> > [3] Xiaoming Lin, Ye Chen, Ziyu Li, and Kang He. Revisiting stock alpha mining
> > based on genetic algorithm. Technical report, Huatai Securities Research Center, 2019b.

---

> > > ### Author Response · Authors · 2024-11-23
> > > **Kindly Requesting Feedback on Our Response**
> > >
> > > Dear Reviewer,
> > >
> > > Thank you very much for your valuable feedback and the time you’ve dedicated to reviewing our work. We genuinely appreciate your detailed and thoughtful comments, which have been immensely helpful in improving our paper.
> > >
> > > We have carefully addressed each of your comments in our response, and we would be very grateful if you could review our responses. If there are any additional questions or concerns, we would be more than happy to address them.
> > >
> > > Thank you once again for your constructive feedback and for helping us enhance the quality of our work.
> > >
> > > Best regards,
> > >
> > > The Authors

---

> > > > ### Comment · Reviewer_JqfW · 2024-11-25
> > > > **Thanks for the thoughtful response**
> > > >
> > > > Thanks to the authors for your thoughtful responses. After reading the other reviews, and the responses, I believe there are significant challenges with the current approach. Specifically, the authors claim a general result, but focus the demonstration on finance. Similarly, although the authors claim generality, the impact of the work appears to be primarily (or entirely?) for financial domains. I do not believe the paper in its current form is suitable for a general machine learning conference such as ICLR.

---

> > > > > ### Author Response · Authors · 2024-11-25
> > > > > **Further Response for Reviewer JqfW**
> > > > >
> > > > > Thank you for your further feedback and for taking the time to review our responses. We appreciate your concerns regarding the generalizability of our work beyond the financial domain.
> > > > >
> > > > > We understand your point that our focus on a financial application may seem to limit the suitability of our paper for ICLR. However, our work aims to contribute not only to RL algorithm development but also to demonstrate its practical application in finance. As indicated in the [Call for Papers of ICLR](https://iclr.cc/Conferences/2025/CallForPapers), ICLR explicitly welcomes applications in diverse domains, including **economics**:
> > > > >
> > > > > ```
> > > > > Subject Areas
> > > > > We consider a broad range of subject areas including feature learning, metric learning, compositional modeling, structured prediction, reinforcement learning, uncertainty quantification and issues regarding large-scale learning and non-convex optimization, as well as applications in vision, audio, speech, language, music, robotics, games, healthcare, biology, sustainability, economics, ethical considerations in ML, and others.
> > > > > ```
> > > > >
> > > > > Importantly, our proposed algorithm is designed with broader applicability in mind, extending to environments characterized by non-stationarity and reward sparsity, challenges that are not unique to finance. While our financial experiments serve as a motivating application, the underlying contributions are relevant to many domains facing similar challenges.
> > > > >
> > > > > We hope this addresses your concerns, and we are open to further suggestions on how we can better convey the general implications of our work.

---

### Official Review · Reviewer_CAKR · 2024-11-01

**Soundness:** 3
**Presentation:** 2
**Contribution:** 2
**Rating:** 5
**Confidence:** 2

**Summary:**

The paper proposes a novel approach combining Distributional Reinforcement Learning (DRL) and quantiled conditional moments (QCMs) to derive synergistic formulaic alphas. The alpha-mining problem is important in finance, and the proposed DRL algorithm could be adaptive for non-stationary and reward-sparse environments. Tests on multiple real-world datasets benchmarked against baseline methods highlight the model's improved performance.

**Strengths:**

The paper is well-organized and generally accessible, with clear explanations of most components used in the proposed methods. Notably, this work is among the first to apply Distributional Reinforcement Learning to the alpha-mining problem. Experimental results demonstrate that the proposed method consistently outperforms all baseline models.

**Weaknesses:**

1. The primary distinction between this work and AlphaGen lies in the use of a Distributional RL algorithm rather than a Proximal Policy Optimization (PPO) approach. The authors argue that their DRL algorithm outperforms PPO by better handling non-stationary and reward-sparse environments, which they claim accounts for the observed performance improvement over AlphaGen. However, the paper lacks empirical evidence, explanation, or references to substantiate this claim. The improvement could also stem from factors such as network parameter sizes or other intrinsic properties of the alpha-mining environment.

2. Certain parts of the paper lack clarity. For instance, while “Distributional Reinforcement Learning” appears in the title, it is not explicitly referenced in the abstract or introduction; instead, it is only implicitly referenced via terms such as "quantile" and the IQN algorithm, which may lead to confusion.

3. There is also a problem in the design of the algorithm. According to Equation (5), the algorithm aims to encourage exploration by incorporating $h$ as the variance of $Q$. The UCB algorithm, for example, used in Chen et al. (2017) [1], leverages a Q-network ensemble variance to guide exploration, aligning with UCB theory as the ensemble variance correlates with the estimation error of the Q function. In contrast, if I understand correctly, the variance in this work represents the intrinsic variance of the $Z$ function, which may not directly correlate with the Q-estimation error.


[1] Chen, Richard Y., et al. "Ucb exploration via q-ensembles." *arXiv preprint arXiv:1706.01502* (2017).

**Questions:**

1. Do the authors have evidence or references supporting that the improved performance is due to the DRL algorithm’s superior handling of non-stationary and reward-sparse environments compared to PPO?

2. In the appendix, it is mentioned that hyperparameters were kept consistent with AlphaGen for a fair comparison. However, given that the PPO policy network in AlphaGen is not directly comparable with the Q-network and quantile network in this paper, could the authors clarify the number of networks and parameters used in both their method and the AlphaGen baseline?

3. In Section 3.3.2, only $\hat{h}$ is estimated from the quantile network. Does this mean that the parameters $s$, $k$, and $\zeta$ discussed in Section 3.3.1 are not utilized in the proposed algorithm? If so, would it be possible to reduce Section 3.3.1 to improve conciseness?

4. According to Equation (5), the algorithm aims to encourage exploration by incorporating $h$ as the variance of $Q$. Are there references supporting this approach? As mentioned in the weakness part, the variance in your work represents the intrinsic variance of the $Z$ function, which may not directly correlate with the Q-estimation error. How does adding $h$ enhance exploration in this context?


[1] Chen, Richard Y., et al. "Ucb exploration via q-ensembles." *arXiv preprint arXiv:1706.01502* (2017).

---

> ### Author Response · Authors · 2024-11-20
> **Response for Reviewer CAKR**
>
> Thank you very much for your insightful comments and suggestions, which give us a great help to improve our article in this revised version. We hope you will find that our revised version has successfully addressed all issues you have raised. Our replies to your questions are as follows.
>
> **Q1: Do the authors have evidence or references supporting that the improved performance is due to the DRL algorithm’s superior handling of non-stationary and reward-sparse environments compared to PPO?**
>
> A: Thank you for your thoughtful question. Yes, our experimental results in Tables 1--2 demonstrate that the improved performance is due to the DRL algorithm's superior handling of non-stationary and reward-sparse environments compared to PPO. The detailed explanations are given below:
>
> First, as highlighted in this paper, we **regard the unbiased QCM variance estimator $\hat{h}$ as the key point of addressing the non-stationary and reward-sparse environments**, where the mean and quantiles estimates may be heavily biased.
>
> Second, we show in Table 1 that the AlphaQCM method significantly outperforms the AlphaGen method, indicating that **the IQN algorithm with the QCM variance estimator $\hat{h}$ beats the PPO algorithm** in non-stationary and reward-sparse environments.
>
> Third, we illustrate in Table 2 that the IQN algorithm with a vanilla variance estimator performs worse than the PPO algorithm on the CSI300 and CSI500 datasets. Although **the IQN algorithm with the vanilla variance estimator employs both the Q and quantile networks, it still underperforms compared to PPO.** This demonstrates that the key factor is the unbiased QCM variance estimator $\hat{h}$, rather than the network architecture or other hyperparameters.
>
> Hopefully, our efforts can well relieve your concern now.
>
> **Q2: In the appendix, it is mentioned that hyperparameters were kept consistent with AlphaGen for a fair comparison. However, given that the PPO policy network in AlphaGen is not directly comparable with the Q-network and quantile network in this paper, could the authors clarify the number of networks and parameters used in both their method and the AlphaGen baseline?**
>
> A: Certainly. The AlphaGen method utilizes two types of networks: an actor network and a critic network, with a total of 298,609 trainable parameters. In contrast, the AlphaQCM method employs four networks: an online Q-network, an online quantile network, and their corresponding target networks (with frozen parameters). The total number of trainable parameters in AlphaQCM is 572,000.
>
> Hopefully, our response can well address your question.
>
> **Q3: In Section 3.3.2, only $\hat{h}$ is estimated from the quantile network. Does this mean that the parameters $s$, $k$ and $\zeta$ discussed in Section 3.3.1 are not utilized in the proposed algorithm? If so, would it be possible to reduce Section 3.3.1 to improve conciseness?**
>
> A: Thank you for your insightful feedback. The parameters $s$ (skewness) and $k$ (kurtosis) are essential for ensuring an unbiased estimator of $h$, as ignoring skewness and kurtosis could result in an incorrect regression model in Equation (4). However, $\zeta$ is not directly used in our algorithm; it is included to illustrate the biased phenomena caused by non-stationarity.
>
> In the revised manuscript, we have streamlined Section 3.3.1 to make it more concise and clear while retaining the necessary context for understanding the role of $s$, $k$, and $\zeta$.
>
> Hopefully, you are satisfied with these changes.
>
> **Q4: According to Equation (5), the algorithm aims to encourage exploration by incorporating $h$ as the variance of $Q$. Are there references supporting this approach? As mentioned in the weakness part, the variance in your work represents the intrinsic variance of the $Z$ function, which may not directly correlate with the Q-estimation error. How does adding enhance exploration in this context?**
>
> A: Thank you for your great question. Yes, there are references supporting the use of the variance of $Z$ as a bonus for efficient exploration. Studies such as [1], [2], [3], and many others demonstrate that using the variance of $Z$ as exploration bonus effectively enhances learning efficiency in reward-sparse environments. While UCB-type bonuses are widely recognized as effective, the $Z$-variance-based exploration represents a distinct approach. Notably, our work is the first one to apply this idea in a non-stationary environment.
>
> This point is mentioned in Line 316 of both the original and revised manuscript. Hopefully, you are satisfied with our efforts.

---

> > ### Author Response · Authors · 2024-11-20
> >
> > **References:**
> >
> > [1] Borislav Mavrin, Hengshuai Yao, Linglong Kong, Kaiwen Wu, and Yaoliang Yu. Distributional reinforcement
> > learning for efficient exploration. In Proceedings of the 36th International Conference on Machine Learning, volume 97, pp. 4424–4434. PMLR, 2019.
> >
> > [2] Zhou, Fan, Jianing Wang, and Xingdong Feng. Non-crossing quantile regression for distributional reinforcement learning. In Advances in neural information processing systems, volume 33, pp. 15909-15919, 2020.
> >
> > [3] Jianye Hao, Tianpei Yang, Hongyao Tang, Chenjia Bai, Jinyi Liu, Zhaopeng Meng, Peng Liu, and
> > Zhen Wang. Exploration in deep reinforcement learning: From single-agent to multiagent domain.
> > IEEE Transactions on Neural Networks and Learning Systems, 35(7):8762–8782, 2024.

---

> > ### Comment · Reviewer_CAKR · 2024-11-20
> >
> > Thanks for the authors' response.
> >
> > Q1: My main concern is whether the three points are enough to conclude that this algorithm handles non-stationary and reward-sparse environments better. The most crucial reason that can't be denied and mentioned by other reviewers is that the environment is very specialized for alpha mining, which is not traditionally treated as a test environment, so people cannot easily see the underlying connections. Furthermore, the performance comparisons among AlphaQCM with vanila variance, AlphaQCM and AlphaGen do not directly support AlphaQCM handling non-stationary and reward-sparse environments better. AlphaQCM is better than AlphaGen, which could be caused by the number of parameters, as I mentioned in Q2. The vanila variance method is not as good as AlphaGen, which could be caused by the complexity of training two networks. These two comparisons can only show that the QCM bonus is necessary for improving the distributional reinforcement learning algorithm. Otherwise, AlphaQCM can't beat AlphaGen.
> >
> > Q2: It is clearer for me now. This should also be included in the appendix.
> >
> > Q3: I know why you need those notations, but QCM is not a new technique developed in this paper, so it needs to be streamlined as long as it is not an obstacle to presenting your main algorithm. In this way, there will be more space to discuss your contributions, like algorithm design or experiments.
> >
> > Q4: Thank you for the clarification. I am not very familiar with distributional RL and missed the bonus term in the references. Though I am still curious how this term helps, which I believe the paper may spend more effort explaining it in their environment, it is clear that the approach is standard in this area.
> >
> > In conclusion, the paper is good and I appreciate the efforts showed in this paper, but it focuses too much on the alpha mining environment. It can't convince me of the general capability to handle non-stationary and reward-sparse environments, nor does the improvement in Alpha mining come from this ability.

---

> > > ### Author Response · Authors · 2024-11-23
> > > **Further Response for Reviewer CAKR (1/2)**
> > >
> > > Thanks for your further comments and suggestions, which motivates us to improve our manuscript again. We hope you will find that our comment and revised version has successfully addressed all issues you have raised.
> > >
> > > **Q1:  My main concern is whether the three points are enough to conclude that this algorithm handles non-stationary and reward-sparse environments better. The most crucial reason that can't be denied and mentioned by other reviewers is that the environment is very specialized for alpha mining, which is not traditionally treated as a test environment, so people cannot easily see the underlying connections. Furthermore, the performance comparisons among AlphaQCM with vanila variance, AlphaQCM and AlphaGen do not directly support AlphaQCM handling non-stationary and reward-sparse environments better. AlphaQCM is better than AlphaGen, which could be caused by the number of parameters, as I mentioned in Q2. The vanila variance method is not as good as AlphaGen, which could be caused by the complexity of training two networks. These two comparisons can only show that the QCM bonus is necessary for improving the distributional reinforcement learning algorithm. Otherwise, AlphaQCM can't beat AlphaGen.**
> > >
> > > A: Thanks for your rigorous explanation. It motivates us to provide an additional experiment to answer your question.
> > >
> > > To determine whether the superior performance of the AlphaQCM method over the AlphaGen method is attributable to its ability to handle the non-stationary and reward-sparse nature of the alpha discovery MDP, rather than simply due to increased model complexity, we designed two new experimental variants:
> > > 1. A larger AlphaGen method with 564,953 trainable parameters, matching the parameter size of the proposed AlphaQCM method.
> > > 2. A smaller AlphaQCM method with 297,070 trainable parameters, aligning with the parameter count of the original AlphaGen method.
> > >
> > > This setup allowed us to isolate the effect of parameter size and focus solely on the algorithmic improvements introduced by AlphaQCM.
> > >
> > > The results are reported in the following table (i.e., Table I.6 in the revised manuscript), where we observed that AlphaQCM consistently outperforms AlphaGen across different parameter scales, showing substantial performance differences. These results indicate that the superior performance of AlphaQCM is not simply a consequence of increased model complexity, but instead reflects its enhanced capability in tackling the non-stationary and reward-sparse alpha discovery MDP.
> > >
> > > |          | CSI300                          |       |     | CSI500 |       |     | Market |       |
> > > | -------- | ------------------------------- | ----- | --- | ------ | ----- | --- | ------ | ----- |
> > > | Method   | Mean                            | Std   |     | Mean   | Std   |     | Mean   | Std   |
> > > |          | Panel A: Small-parameter Method |       |     |        |       |     |        |       |
> > > | AlphaGen | 8.13%                           | 0.94% |     | 8.08%  | 1.23% |     | 6.04%  | 1.78% |
> > > | AlphaQCM | 8.23%                           | 0.90% |     | 9.68%  | 1.07% |     | 9.12%  | 1.47% |
> > > |          | Panel B: Large-parameter Method |       |     |        |       |     |        |       |
> > > | AlphaGen | 8.09%                           | 1.33% |     | 8.40%  | 1.48% |     | 6.67%  | 2.03% |
> > > | AlphaQCM | 8.49%                           | 1.03% |     | 9.55%  | 1.16% |     | 9.16%  | 1.61% |
> > >
> > >
> > > We understand that the specialized nature of the alpha mining environment may make it challenging to generalize our results. However, we hope that this ablation study on parameter size alleviates your concerns and highlights that the performance gains are indeed rooted in the algorithm’s design, not merely in increased complexity.
> > >
> > > Details of the additional experiment have been provided in Appendix I for further review.
> > >
> > > **Q2: It is clearer for me now. This should also be included in the appendix.**
> > >
> > > A: Of course. We have included this part in Appendix I, as suggested. We appreciate your input in helping us improve the clarity of our work.

---

> > > > ### Author Response · Authors · 2024-11-23
> > > > **Further Response for Reviewer CAKR (2/2)**
> > > >
> > > > **Q3: I know why you need those notations, but QCM is not a new technique developed in this paper, so it needs to be streamlined as long as it is not an obstacle to presenting your main algorithm. In this way, there will be more space to discuss your contributions, like algorithm design or experiments.**
> > > >
> > > > A: Thank you for your valuable suggestion. We understand your point regarding streamlining the discussion of QCM to make more space for our main contributions.
> > > >
> > > > Since we are still in the discussion phase, we try to avoid making a large amount of modifications at this stage. However, we fully intend to revise this section to be more concise once the discussion concludes, ensuring that the focus remains on the main contributions, such as the algorithm design and experimental insights.
> > > >
> > > > **Q4: Thank you for the clarification. I am not very familiar with distributional RL and missed the bonus term in the references. Though I am still curious how this term helps, which I believe the paper may spend more effort explaining it in their environment, it is clear that the approach is standard in this area.**
> > > >
> > > > A: Thank you for your understanding and for pointing out the need for further clarification.
> > > >
> > > > As explained in [1], the variance of the $Z$ function captures both parametric and intrinsic uncertainties, which can be directly linked to the challenges of non-stationarity and reward-sparsity in our environment. Specifically, the exploration bonus term, $\hat{h}$, encourages the agent to explore the states with the highest uncertainty. This exploration behavior effectively drives the agent towards informative experiences that are crucial for overcoming non-stationarity and reward sparsity.
> > > >
> > > > By focusing on these informative experiences, our approach mitigates the negative impacts associated with reward sparsity and non-stationarity, ultimately enabling the agent to learn efficiently in a dynamic environment. We hope this additional detail helps clarify the importance of the bonus term and its role in the overall learning process.
> > > >
> > > > We appreciate your interest and hope our efforts to clarify this aspect have been helpful.

---

### Official Review · Reviewer_Ksp7 · 2024-11-03

**Soundness:** 2
**Presentation:** 1
**Contribution:** 1
**Rating:** 3
**Confidence:** 4

**Summary:**

This paper introduces AlphaQCM, a distributional reinforcement learning method designed for the discovery of  the so-called formulaic alphas in financial markets, with a focus on overcoming the challenges of non-stationarity and reward sparsity inherent in alpha-mining. The authors conceptualize alpha discovery as a non-stationary and reward-sparse MDP and employ the quantiled conditional moments (QCM) method to estimate unbiased variances as exploration bonuses. This setup enables efficient exploration of the vast search space in formulaic alpha generation. The paper further demonstrates that AlphaQCM outperforms existing alpha-mining methods, such as AlphaGen and genetic programming, across multiple financial datasets.

**Strengths:**

The strength of this paper include the development of a new RL-based framework that could potentially address the mentioned financial problem.

**Weaknesses:**

I believe this paper has several fundamental weaknesses.

First, I find the paper poorly organized and hard to read. This includes several aspects:

1. The authors frequently refer to other sources for definitions of financial terminologies. Given the niche nature of this topic, many researchers are unlikely to be familiar with the setup, and they may not consult external references to understand it fully. As a result, even after multiple readings of Figure 1, I am still unclear on the role of certain tokens.

2. As an expert in theoretical reinforcement learning, I struggled to understand the setup after reading Section 3.2 multiple times. The paper lacks a foundational formulations of the underlying MDP (For example, I didn't see a formal definition of the transition kernel, let alone an explanation of the non-stationarity issue). In my opinion, this significantly affects the paper’s readability and clarity.

Second, I feel that this paper mainly applies off-the-shelf RL algorithms to a specific financial application. Furthermore, it offers very little insight into why RL is particularly beneficial in this setting or how it advances researchers' understanding of the underlying problems. In other words, the paper lacks depth.

Third, the experiments rely solely on data from the Chinese market, which falls outside the traditional scope of empirical studies in finance, financial economics, or asset pricing. While using Chinese financial data is not inherently problematic, including comparisons with well-established financial datasets would aid in communicating the findings, as researchers could more easily relate the results to their existing understanding. Given that many of these datasets are publicly available, this additional comparison would be straightforward to implement.

**Questions:**

See Weaknesses.

---

> ### Author Response · Authors · 2024-11-20
> **Response for Reviewer Ksp7**
>
> Thank you for your thoughtful comments. To enhance the quality of this paper, we have revised the manuscript in accordance with your suggestions. Below, we provide a point-by-point response to your questions:
>
> **Q1: The authors frequently refer to other sources for definitions of financial terminologies. Given the niche nature of this topic, many researchers are unlikely to be familiar with the setup, and they may not consult external references to understand it fully. As a result, even after multiple readings of Figure 1, I am still unclear on the role of certain tokens.**
>
> A: Thank you for your insightful comments. To improve readability and clarity of this paper for CS researchers, we have revised the manuscript, particularly the introduction section.
>
> Due to space constraints, we only provide definitions for a few token examples in Lines 188–193 of the revised manuscript. The complete details for all tokens are included in Table B.3 for reference.
>
> Hopefully, you are satisfied with our efforts.
>
> **Q2:As an expert in theoretical reinforcement learning, I struggled to understand the setup after reading Section 3.2 multiple times. The paper lacks a foundational formulations of the underlying MDP (For example, I didn't see a formal definition of the transition kernel, let alone an explanation of the non-stationarity issue). In my opinion, this significantly affects the paper’s readability and clarity.**
>
> A: Thank you for your valuable feedback. To clarify the specification of the MDP, we have revised Section 3.2 to improve its clarity. In particular, we have explicitly defined the transition kernel in the alpha discovery MDP and provided additional explanation in Lines 230–233 of the revised manuscript.
>
> Hopefully, you are satisfied with these changes.
>
> **Q3: I feel that this paper mainly applies off-the-shelf RL algorithms to a specific financial application. Furthermore, it offers very little insight into why RL is particularly beneficial in this setting or how it advances researchers' understanding of the underlying problems. In other words, the paper lacks depth.**
>
> A: Thank you for your insightful feedback. In our revised manuscript, we emphasize the following key points to address these concerns:
>
> 1. *Why RL is particularly beneficial in this setting* : As outlined in the manuscript, RL methods outperform existing approaches in several key aspects:
>       * Classical empirical methods struggle to identify subtle and intricate alphas influenced by investor behavior and psychology.
>       * Machine learning models, while powerful, are often too complex to provide a formulaic expression, which can result in trust issues.
>       * Genetic programming methods fail in this domain due to the vast search space they must explore.
> RL provides a promising solution to address all these limitations effectively.
>
> 2. *Insights gained from this paper*: By formulating the alpha discovery problem as finding an optimal policy for an MDP, we highlight the challenges of non-stationarity and reward sparsity. These challenges can cause classical RL algorithms to fail in finding optimal policy, leading to suboptimal empirical performance. Our proposed AlphaQCM method addresses these two issues effectively and demonstrates its potential for application to other non-stationary and/or reward-sparse MDPs.
>
> We hope the above explanations better convey the depth of our contributions and illustrate the value of applying RL in this context.
>
> **Q4: The experiments rely solely on data from the Chinese market, which falls outside the traditional scope of empirical studies in finance, financial economics, or asset pricing. While using Chinese financial data is not inherently problematic, including comparisons with well-established financial datasets would aid in communicating the findings, as researchers could more easily relate the results to their existing understanding. Given that many of these datasets are publicly available, this additional comparison would be straightforward to implement.**
>
> A: Thank you for raising this point. Since the AlphaGen method is the closest comparison method, we followed the experimental settings in [1] to ensure a fair comparison. Additionally, in Line 417 of the revised manuscript, we have elaborated on the motivation for using the Chinese stock market dataset.
>
> While we acknowledge the value of including comparisons with well-established financial datasets, our primary focus is to benchmark against AlphaGen under consistent experimental conditions. Future extensions of this work could incorporate more diverse datasets to provide broader generalizability and relevance.
>
>
> **References:**
>
> [1] Shuo Yu, Hongyan Xue, Xiang Ao, Feiyang Pan, Jia He, Dandan Tu, and Qing He. Generating synergistic formulaic alpha collections via reinforcement learning. In Proceedings of the 29th ACM SIGKDD Conference on Knowledge Discovery and Data Mining, KDD ’23, pp. 5476–5486. Association for Computing Machinery, 2023.

---

> > ### Author Response · Authors · 2024-11-23
> > **Request for Feedback on Revised Responses**
> >
> > Dear Reviewer,
> >
> > We greatly appreciate the time and effort you have invested in reviewing our manuscript and providing such insightful feedback. Your thoughtful comments have been instrumental in refining our work.
> >
> > We have carefully revised our paper and prepared responses to each of your comments. We kindly ask if you could review these responses at your convenience. If there are any further questions or aspects you'd like us to clarify, we would be more than happy to continue this discussion.
> >
> > Thank you once again for your valuable feedback and support in enhancing the quality of our paper.
> >
> > Best regards,
> >
> > The Authors

---

> ### Comment · Reviewer_Ksp7 · 2024-11-27
> **2nd Round Response**
>
> Thanks for replying.
>
> In terms of writing, after reading the revisions, I can only find a marginal improvement. I think this paper still need an overhaul to be understandable to the general audience.
>
> I appreciate that the author try to justify the RL approach. But according to the reasoning:
>
> "
> Classical empirical methods struggle to identify subtle and intricate alphas influenced by investor behavior and psychology.
>
> Machine learning models, while powerful, are often too complex to provide a formulaic expression, which can result in trust issues.
>
> Genetic programming methods fail in this domain due to the vast search space they must explore. RL provides a promising solution to address all these limitations effectively.
> "
>
> The authors just list problems of the existing approach, but what are the key elements in RL that make things work? It would be helpful for some ablation studies that could explain the pros and cons for using RL, since this is a machine learning conference paper. Also, similar issues arise for intuitions that why it can address the non-stationary problem, which is generally challenging in the standard RL setting.

---

### Meta-Review · Area_Chair_YtrS · 2024-12-19

**Metareview:**

Summary:
This paper explores the application of reinforcement learning (RL) to alpha discovery, a specific task in finance. Compared to Alphagen, the most closely related prior work, the proposed AlphaQCM method addresses nonstationarity and sparse reward issues. The authors applied AlphaQCM to financial datasets and reported improved results.

Strengths:

- All reviewers agree that AlphaQCM enhances existing baselines by utilizing a QCM method to effectively handle nonstationarity and reward sparsity. The application of RL to less frequently explored domains such as finance is an interesting and worthwhile endeavor.

Weaknesses:

- All reviewers, including myself, recognize that this paper primarily appears to be an application of an existing RL method to a finance problem, without introducing significant algorithmic or technical novelty. While studying RL applications in finance is commendable, the overall contribution of this work feels limited in terms of advancing RL methodologies.

Recommendation:
I recommend rejecting this paper in its current form. The authors are encouraged to either submit their work to a finance-focused conference or revise the manuscript to better highlight any novel technical contributions.

**Additional Comments On Reviewer Discussion:**

The authors provided additional experiments during the discussion phase; however, these updates do not fully address the concerns raised by the reviewers regarding the lack of technical depth and insufficiently addressed challenges.

---

### Decision · Program_Chairs · 2025-01-22

Reject